# Reprogramming of bone marrow myeloid progenitor cells in patients with severe coronary artery disease

**Marlies P Noz[1], Siroon Bekkering[1], Laszlo Groh[1], Tim MJ Nielen[2], Evert JP Lamfers[2], Andreas Schlitzer[3], Saloua El Messaoudi[4], Niels van Royen[4], Erik HJPG Huys[5], Frank WMB Preijers[5], Esther MM Smeets[6], Erik HJG Aarntzen[6], Bowen Zhang[7], Yang Li[1,7], Manita EJ Bremmers[8], Walter JFM van der Velden[8], Harry Dolstra[5], Leo AB Joosten[1,9], Marc E Gomes[2], Mihai G Netea[1,10], Niels P Riksen[1]***

[1]Department of Internal Medicine and Radboud Institute for Molecular Life Science (RIMLS), Radboud University Medical Center, Nijmegen, Netherlands; [2]Department of Cardiology, Canisius Wilhelmina Hospital, Nijmegen, Netherlands; [3]Quantitative Systems Biology, Life & Medical Sciences Institute, University of Bonn, Single Cell Genomics and Epigenomics Unit at the German Center for Neurodegenerative Diseases and the University of Bonn, Bonn, Germany; [4]Department of Cardiology, Radboud University Medical Center, Nijmegen, Netherlands; [5]Department of Laboratory Medicine – Laboratory for Haematology, Radboud University Medical Center, Nijmegen, Netherlands; [6]Department of Radiology and Nuclear Medicine, Radboud University Medical Center, Nijmegen, Netherlands; [7]Department of Computational Biology for Individualised Infection Medicine, Centre for Individualised Infection Medicine (CiiM) & TWINCORE, joint ventures between the Helmholtz-Centre for Infection Research (HZI) and the Hannover Medical School (MHH), Hannover, Germany; [8]Department of Haematology, Radboud University Medical Center, Nijmegen, Netherlands; [9]Department of Medical Genetics, Iuliu Hatieganu University of Medicine and Pharmacy, Cluj-Napoca, Romania; [10]Department for Genomics & Immunoregulation, Life and Medical Sciences Institute (LIMES), University of Bonn, Bonn, Germany

**\*For correspondence:**
niels.riksen@radboudumc.nl

**Competing interests:** The authors declare that no competing interests exist.

**Abstract** Atherosclerosis is the major cause of cardiovascular disease (CVD). Monocyte-derived macrophages are the most abundant immune cells in atherosclerotic plaques. In patients with atherosclerotic CVD, leukocytes have a hyperinflammatory phenotype. We hypothesize that immune cell reprogramming in these patients occurs at the level of myeloid progenitors. We included 13 patients with coronary artery disease due to severe atherosclerosis and 13 subjects without atherosclerosis in an exploratory study. Cytokine production capacity after *ex vivo* stimulation of peripheral blood mononuclear cells (MNCs) and bone marrow MNCs was higher in patients with atherosclerosis. In BM-MNCs this was associated with increased glycolysis and oxidative phosphorylation. The BM composition was skewed towards myelopoiesis and transcriptome analysis of HSC/GMP cell populations revealed enrichment of neutrophil- and monocyte-related pathways. These results show that in patients with atherosclerosis, activation of innate immune cells occurs at the level of myeloid progenitors, which adds exciting opportunities for novel treatment strategies.

## Introduction

Atherosclerotic cardiovascular disease (CVD), including myocardial infarction and stroke, is the leading cause of death worldwide. Major risk factors for CVD include dyslipoproteinemia, smoking, hypertension, obesity, and diabetes. However, despite optimal pharmacological treatment of these risk factors, a considerable residual CVD risk remains. Evidence is rapidly accumulating that chronic low-grade inflammation of the vascular wall is a key pathophysiological component of atherosclerosis, and that treatment with anti-inflammatory drugs, such as canakinumab and colchicine, additionally lowers future CVD risk (*Nidorf et al., 2020*; *Ridker et al., 2017*; *Tardif et al., 2019*). More detailed knowledge of this inflammatory process and the role of individual immune cells would allow the development of more specific and safe anti-inflammatory therapies.

Within atherosclerotic plaques, monocyte-derived macrophages are the most abundant immune cells, which play a key role in initiation, progression, and destabilization of these plaques, although the mechanisms driving the persistent inflammatory activation are poorly understood (*Moore et al., 2013*). In patients with CVD or risk factors for CVD, circulating leukocytes show differences in composition and individual cell phenotypes, compared to healthy individuals. Isolated monocytes from patients with CVD are characterized by an increased cytokine production capacity and higher glycolytic metabolism (*Bekkering et al., 2016*; *Elsenberg et al., 2013*; *Shirai et al., 2016*). This also holds true for patients with increased CVD risk due to familial hypercholesterolemia (*Bekkering et al., 2019*). In addition, higher circulating levels of granulocytes are associated with an increased CVD risk in a general population study (*Fani et al., 2020*).

Experimental studies in animal models of atherosclerosis have revealed reprogramming of hematopoietic stem and progenitor cells (HSPCs) in the bone marrow as a cause of these changes in circulating innate immune cells under conditions that predispose to CVD, including hyperlipidemia (*Christ et al., 2018*), and stress (*Heidt et al., 2014*). Western-type diet feeding of low-density lipoprotein receptor deficient (*Ldlr-/-*) mice induces inflammatory activation of monocytes by epigenetic reprogramming of HSPCs which persists in normocholesterolemic conditions (*Christ et al., 2018*). This resembles the finding of persistent functional and transcriptional hyperresponsive monocytes in patients with dyslipidemia despite cholesterol-lowering treatment (*Bekkering et al., 2019*). In a retrospective analysis of bone marrow samples from cancer patients, HSPCs had a higher proliferative potential in patients with known atherosclerotic CVD than in control patients (*van der Valk et al., 2016*). A detailed assessment, however, of HSPC composition and function in patients with atherosclerosis is lacking.

Here we performed a comprehensive study in patients with and without coronary artery disease (CAD). We obtained circulating leukocytes and HSPCs and assessed composition with flow cytometry, as well as cytokine production capacity, metabolism, and the transcriptional profile. In addition, we performed 2'-deoxy-2'-[$^{18}$F]fluoro-D-glucose positron-emission-tomography ([$^{18}$F]FDG PET/CT) scanning to detect bone marrow and splenic activity. We report that bone marrow mononuclear cells in patients with atherosclerosis are characterized by enhanced cytokine production capacity and an increased metabolic rate. Flow cytometry and RNAseq analysis revealed inflammatory transcriptional reprogramming and myeloid skewing. These results show that in patients with atherosclerosis, activation of the innate immune system occurs at the level of bone marrow myeloid progenitors, which adds exciting opportunities for novel treatment strategies.

## Results

### Group characteristics

Thirteen individuals with CAD due to severe atherosclerosis with a median Total Plaque Score (TPS) of 14[9-15], and thirteen individuals without coronary atherosclerosis (TPS = 0) participated in the study (*Table 1*). As expected, patients with atherosclerosis more often received lipid-lowering therapy with statins, which was associated with a lower total and LDL-cholesterol concentration. All outcomes were corrected for the age and BMI because these are known modulators of innate immune cell function (*Ter Horst et al., 2016*).

**Table 1.** Group characteristics.

| Characteristics | Individuals with CAD (n = 13) | Individuals without atherosclerosis (n = 13) |
|---|---|---|
| Age (years) | 59.8 ± 9.7 | 52.2 ± 10.4 |
| Sex (% men, n) | 100 (13) | 100 (13) |
| BMI (kg/m$^2$) | 27.8 ± 2.8 | 25.8 ± 2.5 |
| SBP (mm Hg) | 133 ± 15 | 126 ± 10 |
| DBP (mm Hg) | 90 ± 8* | 84 ± 6 |
| Hypertension (%, n) | 93 (12)** | 31 (4) |
| Current smoking (%, n) | 23 (3) | 31 (4) |
| Calcium score (HU) | 445 [213-781]** | 0 [0] |
| Total Plaque score (0–16)‡ | 14 [9-15]*** | 0 [0] |
| Lipid-lowering therapy (%, n) | 77 (10)*** | 8 (1) |
| Acetylsalicylic acid use (%, n) | 69 (9)*** | 0 (0) |
| ACE-inhibitor use (%, n) | 23 (3) | 8 (1) |
| β-blocker use (%, n) | 23 (3) | 8 (1) |
| Glucose (mmol/L) | 5.9 ± 0.8 | †5.7 ± 0.7 |
| Creatinine (μmol/L) | 89 ± 14 | 91 ± 16 |
| Tchol (mmol/L) | 4.51 ± 0.86** | 5.61 ± 0.61 |
| LDLc (mmol/L) | †2.52 ± 0.97** | 3.56 ± 0.64 |
| HDLc (mmol/L) | 1.25 ± 0.31 | 1.51 ± 0.34 |
| TG (mmol/L) | 1.98 ± 1.97 | 1.20 ± 0.38 |
| nHDLc (mmol/L) | 3.26 ± 1.06* | 4.11 ± 0.75 |

Data are reported as mean ± SD, as mean (number of participants), or as median [interquartile range] and compared with the appropriate statistical tests. ‡ TPS was calculated for participants with a calcium score of <400 HU (n = 6). † Data is missing for one participant. * indicates p<0.05, **: p<0.01, ***: p<0.001.

## Circulating inflammatory markers

The distribution of immune cells and monocyte subpopulations was similar between groups (*Table 2*). Integrin CD11b expression on monocytes tended to be higher in patients with CAD (p=0.09). Circulating endothelial dysfunction marker E-selectin was higher in patients with atherosclerosis (p<0.05).

## Cytokine production capacity of circulating PBMCs is higher in atherosclerosis

Previously, we showed in a comparable study cohort that the LPS-induced production of IL-6, TNFα, IL-1β, and IL-8 in PBMCs was higher in patients with CAD compared to controls (*Bekkering et al., 2016*). We could confirm this in the current study for IL-8 (p<0.01), with a similar pattern for IL-6 (p=0.08) and TNFα (p=0.13) (*Figure 1A*). There were no significant differences in response to TLR2 agonist Pam3Cys stimulation (*Figure 1B*).

## HSPC composition is changed in patients with CAD

We assessed the HSPC composition using flow cytometry. Although the total percentage of HSPCs was similar between groups, the percentage of multipotent progenitors (HSC/MPP, p<0.05) was higher in patients with CAD (*Figure 2A,B*). Trends for a higher percentage of common myeloid progenitors (CMP, p=0.06) and common lymphoid progenitors (CLP, p=0.08) were observed in patients with CAD (*Figure 2C,D*). In addition, the percentage of pre-monocytes was higher (p<0.05), which was associated with lower megakaryocyte erythrocyte progenitor percentages (MEP, p<0.05) (*Figure 2F,G*). Within the bone marrow, no difference in the percentage of monocytes (3.3 [3.1–4.4]% in patients versus 3.4 [3.0–4.6] in controls), and monocyte subpopulations, that is, classical (87 [82–88] % versus 82 [79-89]), intermediate (7.5 [7.1–10.8]% versus 9.3 [6.1–11.5]) and non-classical monocytes (5.1 [4.2–8.1]% versus 8.1 [3.6–11.4]), was observed. Interestingly, the percentage of

**Table 2.** Circulating immune cells and inflammatory markers in patients and controls.

| Cell types | Individuals with CAD | Individuals without atherosclerosis |
|---|---|---|
| WBC ($10^6$/mL) | †5.5 [4.9–6.0] | 5.4 [4.8–6.7] |
| Neutrophils ($10^6$/mL) | †3.2 [2.5–3.6] | 3.0 [2.6–4.0] |
| Lymphocytes ($10^6$/mL) | 1.7 [1.3–1.9] | 1.8 [1.5–2.5] |
| Monocytes ($10^6$/mL) | 0.53 [0.42–0.67] | 0.55 [0.45–0.66] |
| Monocytes (%) | 9.8 [8.0–11.5] | 9.3 [7.7–11.1] |
| Classical monocytes (%gated) | 78.1 [72.8–80.3] | 72.8 [70.1–85.5] |
| Intermediate monocytes (%gated) | 9.8 [8.2–14.2] | 10.1 [7.6–13.7] |
| Nonclassical monocytes (%gated) | 12.2 [9.3–14.3] | 13.1 [6.3–18.2] |
| CCR2+ monocytes (%gated) | 80.5 [73.0–82.3] | †77.6 [71.4–86.3] |
| CD11b expression monocytes (MFI) | 10490 [7814–12025] | ††6978 [6512–10041] |
| CD41+ monocytes (%gated) | 7.8 [6.5–9.7] | 8.7 [7.6–8.9] |
| Inflammatory markers | | |
| IL-1β (pg/mL) | †0.12 [0.09–0.17] | 0.12 [0.06–0.15] |
| IL-1Ra (pg/mL) | 271 [197-338] | 212 [165-253] |
| IL-6 (pg/mL) | 2.31 [1.37–2.86] | †1.61 [1.23–2.19] |
| IL-18 (pg/mL) | 162 [127-227] | 195 [144-236] |
| hsCRP (pg/mL) | 1.66 [0.83–4.87] | 1.37 [0.53–3.84] |
| E-selectin (ng/mL) | †**11.74 [7.65–15.10]\*** | **8.45 [4.52–14.06]** |
| VCAM-1 (ng/mL) | 773 [711-859] | 769 [643-844] |
| MMP2 (ng/mL) | 354 [264-434] | 341 [250-427] |

Circulating concentrations of cells and inflammatory markers in individuals with CAD (n = 13) compared to individuals without atherosclerosis (n = 13). Median with [IQR]. P-values are corrected for age and BMI with ANCOVA. Outliers were removed with an SD of >2.5 of Z-scores. † Data is missing for one participant. \*p<0.05, \*\*p<0.01. HSPCs: hematopoietic stem and progenitor cells.

circulating multipotent progenitors (HSC/MPP) was higher in patients with CAD (p<0.05) (*Figure 2I*). Similar patterns were found using absolute cell counts of progenitor populations.

## Functional and metabolic reprogramming of bone marrow MNCs in atherosclerosis

The immune response of BM-MNCs was determined after *ex vivo* 24-hour stimulation with TLR agonists. TNFα production after LPS stimulation was higher in patients with atherosclerosis (p<0.05), with similar patterns for IL-6 (p=0.12), IL-8 (p=0.10), and IL-1Ra (p=0.11) although these differences did not reach statistical significance (*Figure 3A*). There were no significant differences in response to Pam3Cys stimulation (*Figure 3B*).

Seahorse respirometry revealed that both the basal and maximal OCR was higher (p<0.01 and p<0.05) in BM-MNCs from patients with CAD compared to control subjects (*Figure 4A,C*), as well as the basal and maximal ECAR (p<0.01 and p<0.05) (*Figure 4B,D*). The measurements were repeated 2 hours after IFN-γ/LPS pre-incubation, in order to assess metabolic function during activation. Again, the maximal OCR (p<0.01) and maximal ECAR (p<0.05) were higher in patients with CAD.

## Proliferation of BM-MNCs

Proliferation assays of BM-MNCs did not reveal significant differences in erythroid-, myeloid-, or granulocyte-macrophage progenitor proliferation rates between patients with CAD and healthy controls (*Figure 5*).

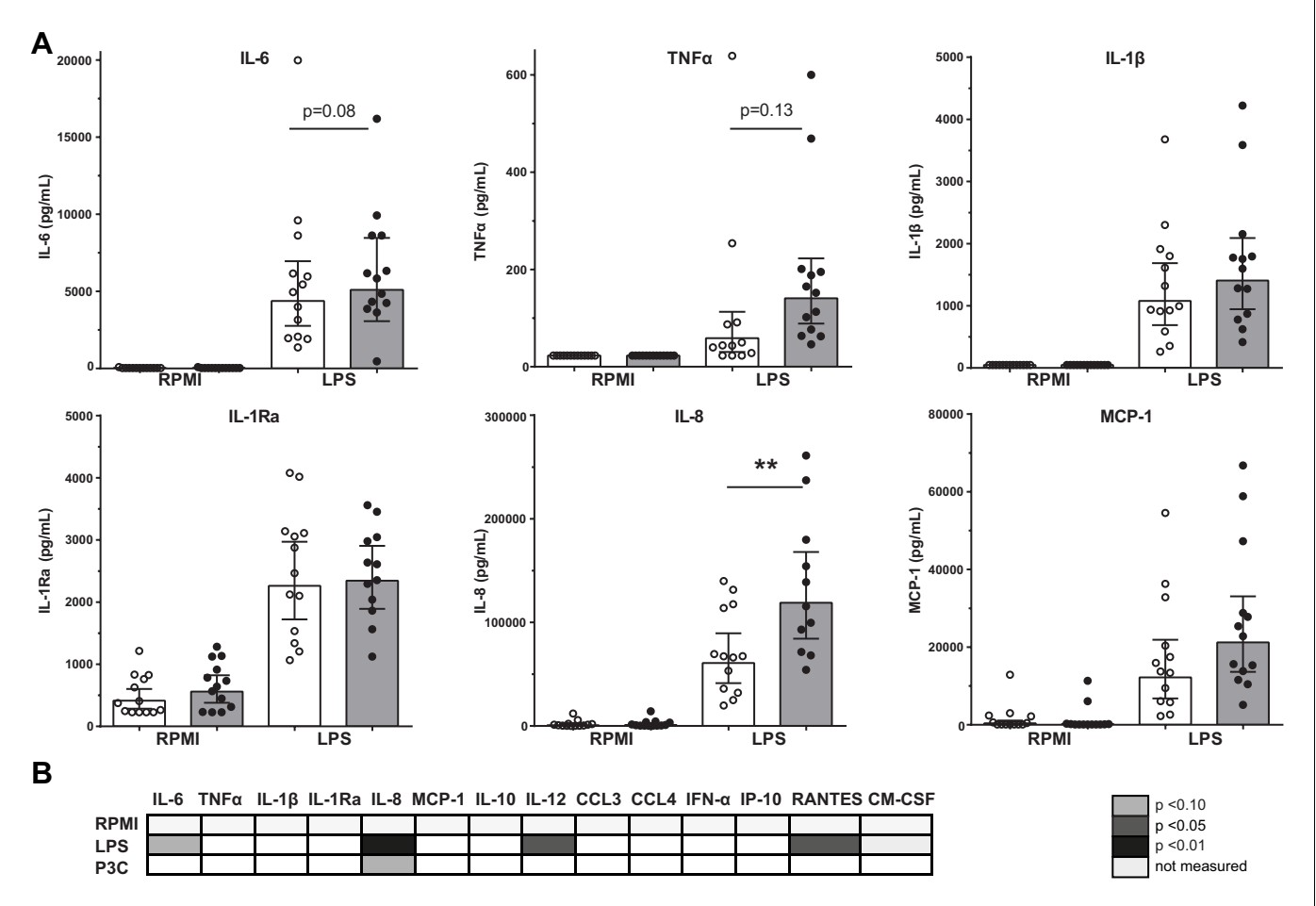

**Figure 1.** Cytokine production capacity of circulating PBMCs. (**A**) Cytokine production capacity of circulating PBMCs after LPS stimulation in control individuals (white bars, n = 13) and individuals with CAD (gray bars, n = 13). Geometric mean with 95% CI. (**B**) Table of cytokine/chemokine production (x-axis) after stimulation with LPS or P3C (y-axis) of PBMCs showing statistical differences between groups. The p-values are corrected for age and BMI with ANCOVA. Outliers were removed with an SD of >2.5 of Z-scores. * indicates p<0.05, **: p<0.01.

## Bone marrow progenitors of patients with CAD are primed to differentiate into an inflammatory myeloid lineage

To further understand the functional changes in HSPCs, we explored the transcriptional signature of HSC, MPP, and GMP cell populations. Differential expression analysis was applied to each population separately and a combined analysis of the populations to identify differentially expressed (DE) genes between patients and control samples. This identified 1747 genes that were differentially regulated in at least one of the cell populations (p≤0.05). A PCA analysis based on DE genes of HSC populations revealed a clear separation between patients and controls (*Figure 6A*). This was less pronounced for the MPPs and GMPs (*Figure 6—figure supplement 1*). Among those DE genes, we observed that four genes were significantly upregulated in the patients compared to the controls in a combined analysis of HSCs, MPPs, and GMPs (*Figure 6B*, p$_{adj}$ <0.1), including *CCR2*, *EPB42*, *FNDC3B*, and *RBMS1*. For individual log fold change (FC) and adjusted p-values, please see *Figure 6—source data 1* and the combined and separated heatmaps for the top 50 DE genes for each population (*Figure 6—figure supplements 2* and *3*). Seven genes were differentially downregulated in the patients, including *PFKP*, *CCDC163P*, *ARMCX4*, *PTK7*, *WDR90*, *ROBO3*, and *FAM84B*. Within the HSC population, only *PROK2* showed a significant upregulation (*Figure 6—figure supplement 3*), whereas in the MPPs and GMPs no genes showed significantly differential upregulation (*Figure 6—figure supplement 3*).

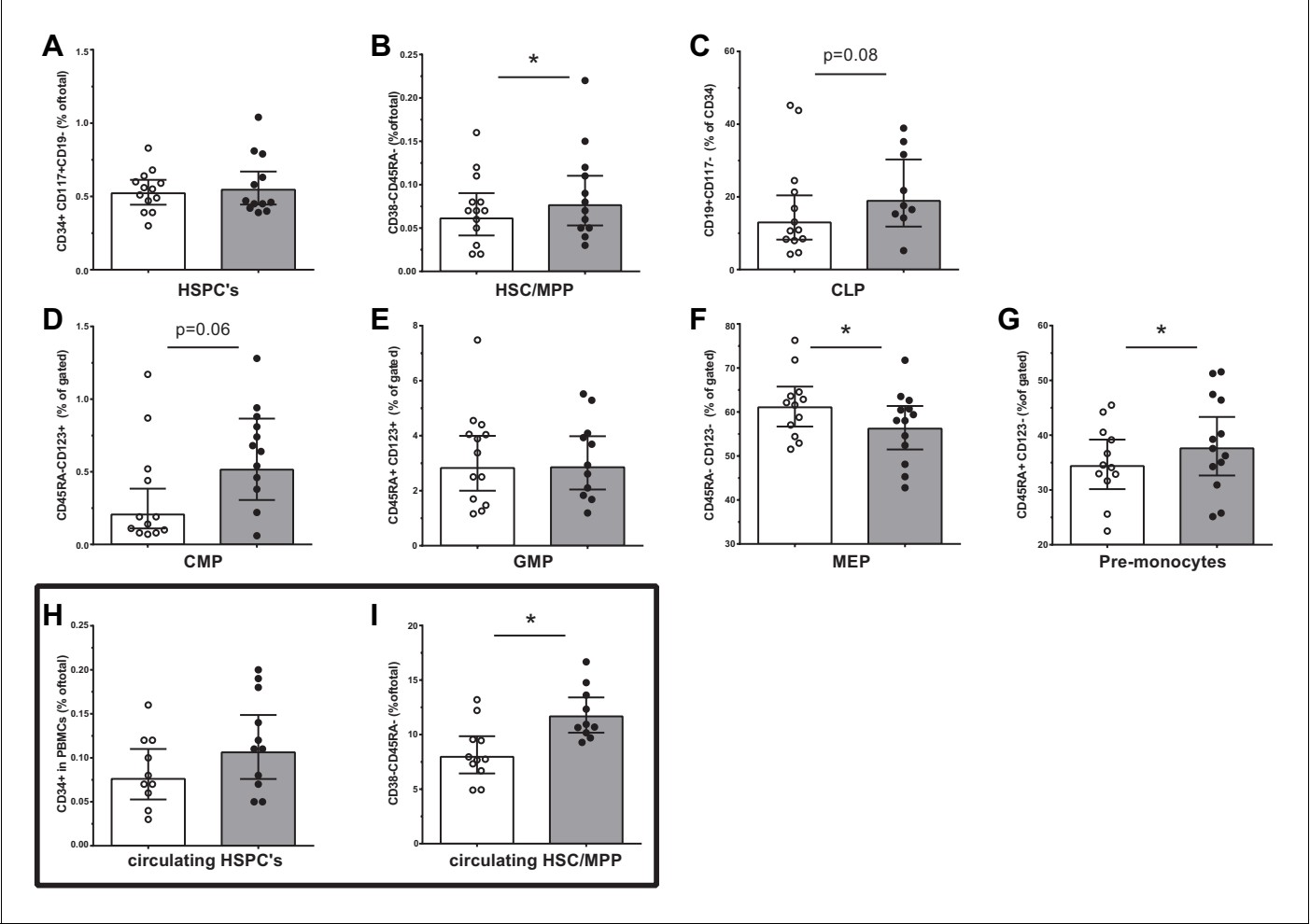

**Figure 2.** Progenitor cell populations in the bone marrow compartment (**A–G**) and in the circulation (**H and I**). Control individuals (white bars, n = 13) and individuals with CAD (gray bars, n = 13). HSC and MPP cell populations were combined as the CD90 expression marker was not available for n = 6 in each study group. Geometric mean with 95% CI. The p-values are corrected for age and BMI with ANCOVA. * indicates p<0.05, **: p<0.01. In top-down order: HSC indicates hematopoietic stem cell, MPP: multipotent progenitor, CLP: common lymphoid progenitor, CMP: common myeloid progenitor, GMP: granulocyte-macrophage progenitor, MEP: megakaryocyte erythrocyte progenitor.

The online version of this article includes the following figure supplement(s) for figure 2:

**Figure supplement 1.** Gating strategy of hematopoietic stem and progenitor cells in the bone marrow.

Subsequently, we performed pathway enrichment analyses on disease-specific DE genes. The differentially upregulated genes of HSCs (p<0.05) showed significant enrichment for neutrophil and monocyte biological processes. These include neutrophil activation pathways, cytokine production pathways, and macrophage activation pathways (FDR <0.05) (*Figure 6C*). Transcriptional signatures of the genes enriched in the neutrophil-related pathways indicated upregulation of genes involved in the development and activation of neutrophils, such as insulin like growth factor two receptor (*IGF2R*), *S100A11*, and *TNFRSF1B* (*Figure 6—source data 2*). Additionally, there was an upregulation of signalling genes in the myeloid lineage, such as *CCR2*, innate immune signal transduction adaptor (*MYD88*), *IL1RN*, *IL18R1*, and Toll-like receptors *TLR2* and *TLR4*. The differentially upregulated genes in the GMPs showed a similar enrichment for neutrophil-related pathways, and pathways related to myeloid cell differentiation and migration (*Figure 6C*). Upregulated genes in myeloid cell differentiation and regulation of hematopoiesis pathways revealed upregulation of important signalling and transcription factors, such as colony stimulating factor three receptor (*CSF3R*), NFKB inhibitor alpha (*NFKBIA*), signal transducer and activator of transcription 1 (*STAT1*), *STAT3,* and

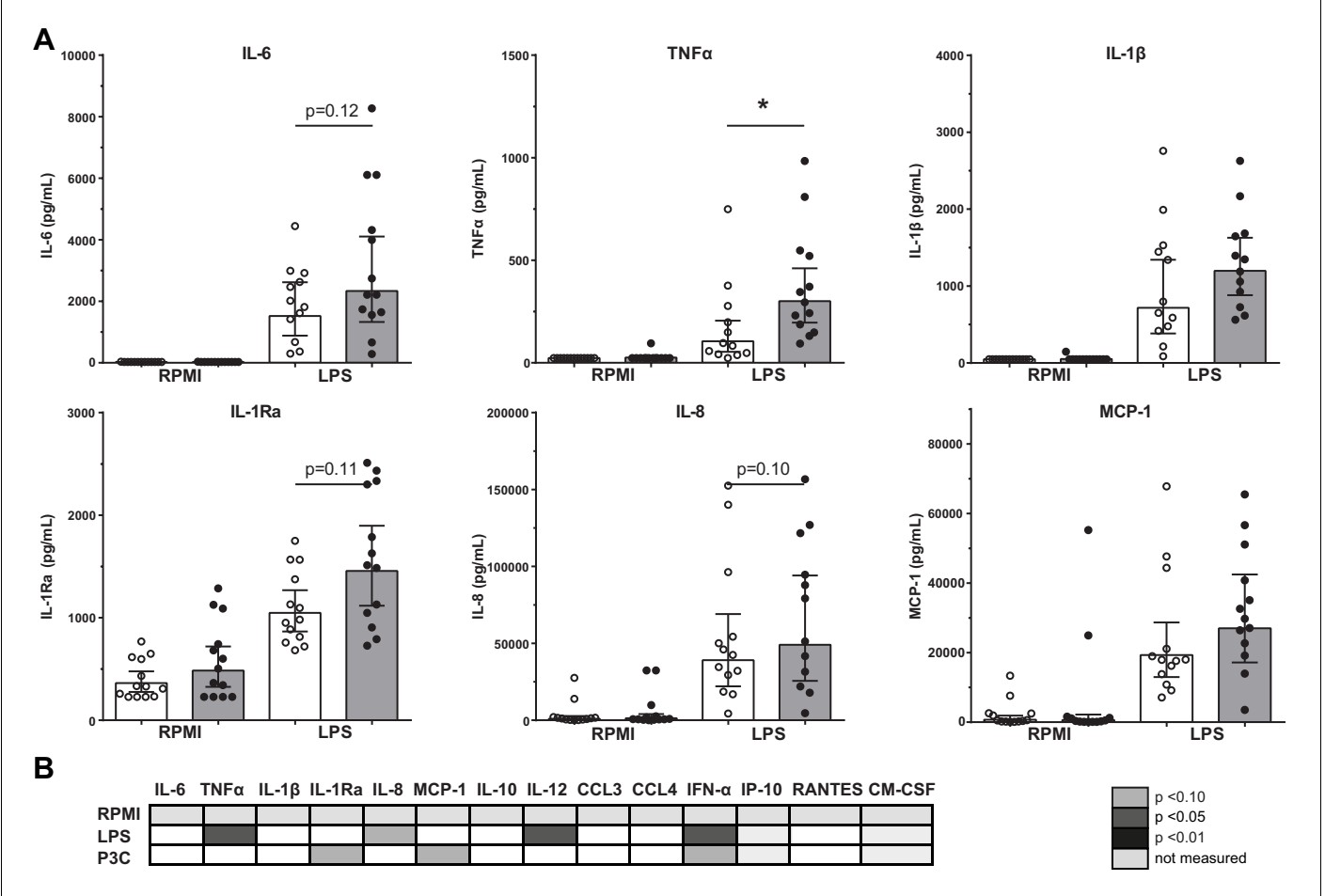

**Figure 3.** Cytokine production capacity of bone marrow MNCs. (**A**) Cytokine production capacity of BM-MNCs after LPS stimulation in control individuals (white bars, n = 13) and individuals with CAD (gray bars, n = 13). Geometric mean with 95% CI. (**B**) Table of cytokine/chemokine production (x-axis) after stimulation with LPS or P3C (y-axis) of BM-MNCs showing statistical differences between groups. The p-values are corrected for age and BMI with ANCOVA. Outliers were removed with an SD of >2.5 of Z-scores. * indicates p<0.05, **: p<0.01.

transforming growth factor beta 1 (*TGFB1*). Repeating these analyses using p<0.01 showed that the observed enriched terms were robust (FDR <0.05).

RNA-seq data have been deposited in the ArrayExpress database at EMBL-EBI (www.ebi.ac.uk/arrayexpress) under accession number E-MTAB-9399.

## Vascular wall inflammation and hematopoietic tissue activation measured with [18F]FDG PET/CT is not higher in patients with CAD

Vascular wall inflammation and hematopoietic tissue activity, as determined by [18F]FDG PET/CT-scan, was similar between patients with CAD and individuals without atherosclerosis (*Figure 7*). Correction for age, BMI, and glucose concentrations did not influence the results.

Interestingly, although the splenic activity was not significantly higher in patients with CAD, the splenic [18F]FDG-uptake correlated with HSPCs and with circulating immune cells (*Figure 7—figure supplement 1*). Splenic [18F]FDG-uptake correlated positively with CCR2 expression on HSPCs ($r_s$ = 0.604, p<0.01), with GMPs ($r_s$ = 0.405, p<0.05), and with circulating leukocyte ($r_s$ = 0.393, p<0.05) and monocyte counts ($r_s$ = 0.588, p<0.01).

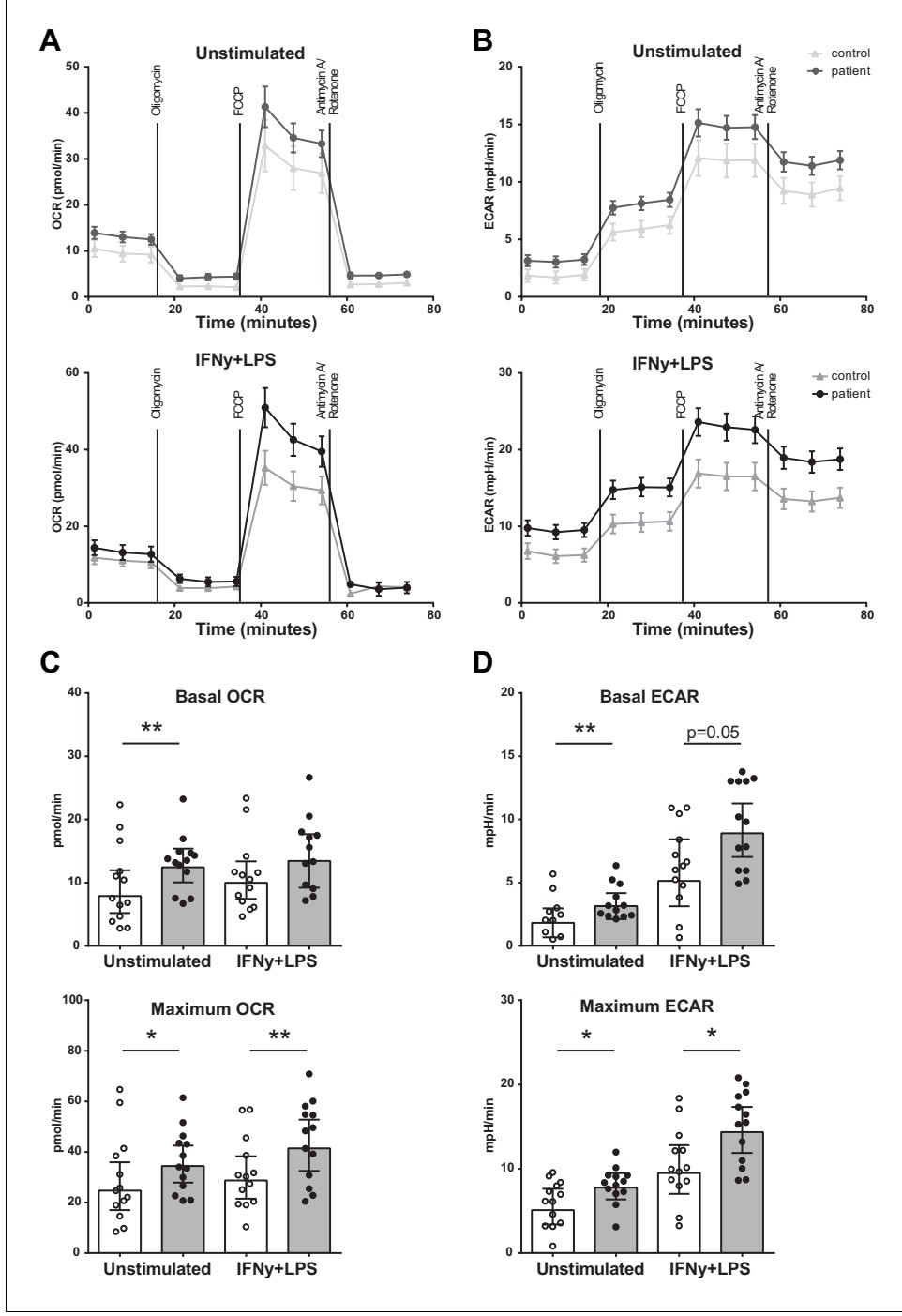

**Figure 4.** Metabolism of BM-MNCs assessed with Seahorse respirometry in unstimulated condition and 2 hours after IFN-γ+LPS stimulation. (A, B) Oxygen consumption and extracellular acidification rates over time using treatment with Oligomycin, FCCP, and Rotenone/Antimycin A. (C, D) Bar graphs of control individuals (white bars, n = 13) and individuals with CAD (gray bars, n = 13). Geometric mean with 95% CI. The p-values are corrected for age and BMI with ANCOVA. * indicates p<0.05, **: p<0.01. IFN-γ+LPS: 2 hr IFN-γ and LPS stimulation.

## Discussion

The role of activated innate immune cells in the development of atherosclerotic plaques is well-established. In the current paper we add an important dimension to this pathophysiological framework by showing for the first time that inflammatory reprogramming of innate immune cells in

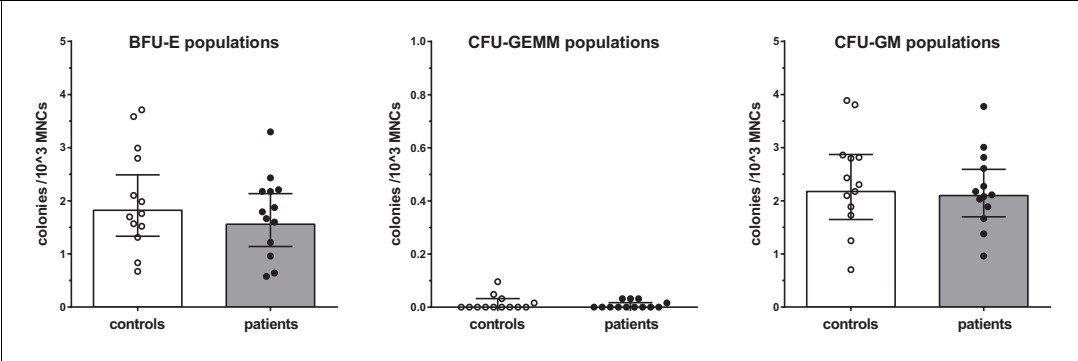

**Figure 5.** Proliferation capacity of bone marrow MNCs. Counted colonies per $10^3$ cultured BM-MNCs of control individuals (white bars, n = 13) and individuals with CAD (gray bars, n = 13). Geometric mean with 95% CI. The p-values are corrected for age and BMI with ANCOVA. BFU-E indicates erythroid progenitor population, CFU-GEMM: myeloid progenitor population, CFU-GM: granulocyte-macrophage progenitor population.

humans with coronary artery disease occurs at the level of HSPCs in the bone marrow. In patients with chronic CAD due to severe coronary atherosclerosis, isolated bone marrow mononuclear cells showed a higher cytokine production capacity and increased metabolic rate compared to individuals without coronary atherosclerosis. Transcriptional signatures of HSPCs were enriched for neutrophil and monocyte related pathways, indicating that the HSPCs are primed to differentiate into an inflammatory myeloid lineage. These findings addinclude an additional layer of inflammatory regulation in atherogenesis which potentially offers novel targets for pharmacological strategies to prevent or treat atherosclerotic CVD.

Our findings align with previous studies in animal models in atherosclerosis that show increased myelopoiesis in specific conditions that promote CVD. Accumulation of cholesterol in HSPC increases proliferation and mobilization from the bone marrow and a differentiation bias towards the myeloid lineage, which is associated with accelerated atherosclerosis (*Yvan-Charvet et al., 2010*). In an animal model of diabetes, higher circulating numbers of inflammatory Ly6-C^hi monocytes and neutrophils are present, which is due to increased myelopoiesis. In this model, the increased myelopoiesis is driven by the increased expression of *S100A8* and *S100A9* in neutrophils (*Nagareddy et al., 2013*). Interestingly, the expression of one of the S100 proteins (*S100A11*) was also increased in the HSCs in patients with CAD in our study. Chronic psychological stress and disturbed sleep have also been reported to increase myelopoiesis, which is associated with elevated circulating levels of neutrophils and Ly6C^high monocytes and augmented atherosclerosis compared with control mice (*Heidt et al., 2014*; *McAlpine et al., 2019*).

Since we did not include patients with specific cardiovascular risk factors, but rather with established symptomatic coronary atherosclerosis, we cannot conclude which factors are responsible for the HSPC reprogramming in our patients. BMI and blood pressure were slightly higher in the patients compared to the controls. Cholesterol concentrations were lower at the moment of inclusion in our study, but might well have been higher previously, since the patients also used more statins. We do not have information about stress and sleep patterns. Another mechanism that might contribute to the HSPC functional reprogramming is trained immunity. Trained immunity describes the phenomenon that brief stimulation of innate immune cells leads to the development of a long-term hyperresponsive phenotype. This is mediated by profound intracellular metabolic and epigenetic reprogramming and occurs both at the level of mature circulating monocytes as well as at the level of their bone marrow progenitors (*Netea et al., 2020*). Christ et al have recently described that a 4-week period of Western-type diet feeding in atherosclerosis prone *Ldlr-/-*mice induced a functional, transcriptional and epigenetic reprogramming of circulating myeloid cells and their bone marrow progenitors, which persisted for at least for weeks after switching back to a chow diet (*Christ et al., 2018*). This was associated with increased circulating concentrations of inflammatory monocytes and granulocytes. Functional enrichment analysis of GMPs showed overrepresentation of TNF and Toll-like receptor signaling pathways, which align with our observations. In contrast, although monocytic signature genes were enriched, most of the granulocytic signature genes were downregulated, including S100A8 and S100A9, in GMPs isolated from WD-fed as compared to CD-

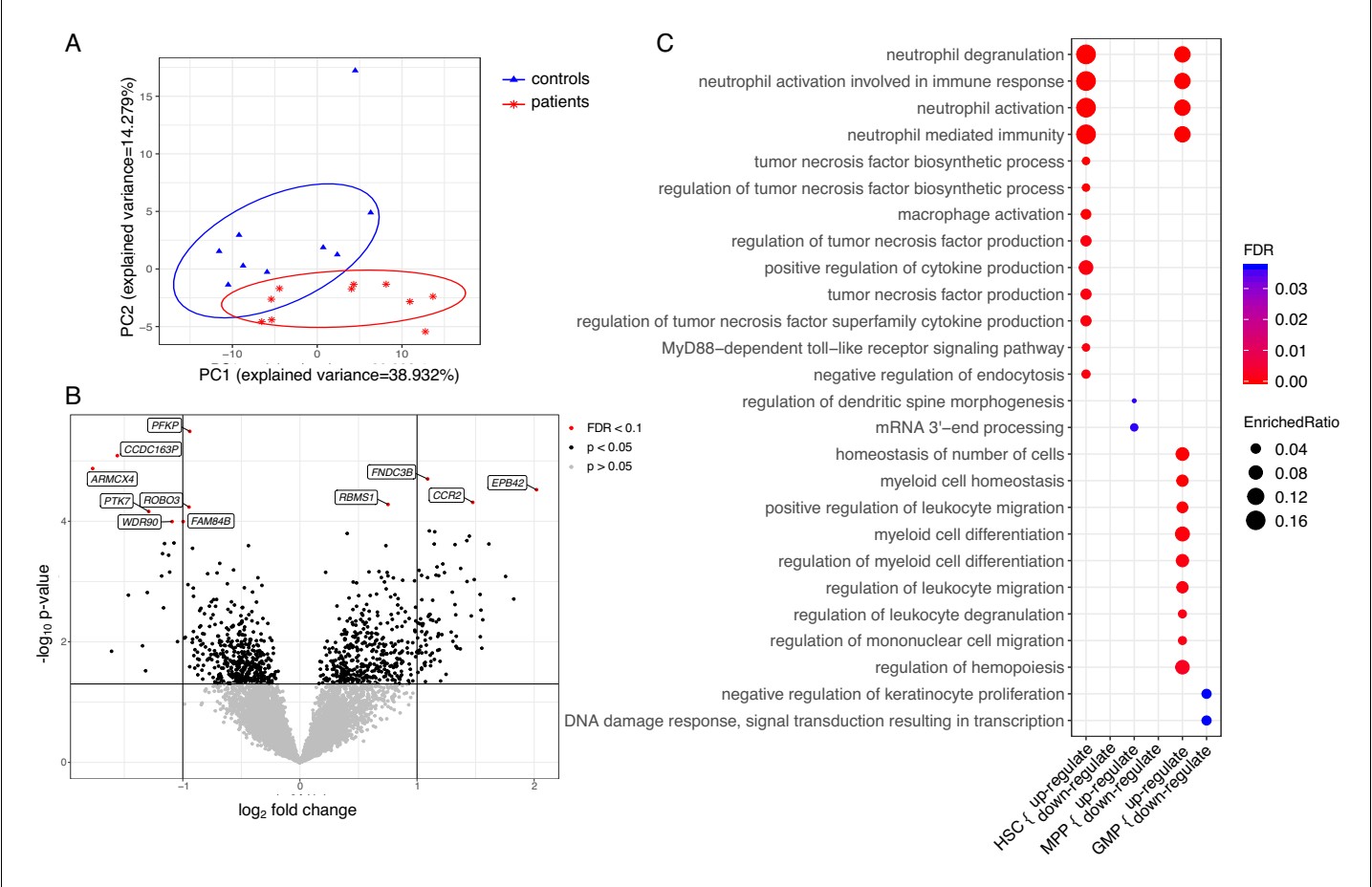

**Figure 6.** Transcriptome analyses of HSC, MPP, and GMP populations. Control individuals (n = 10) versus individuals with CAD (n = 10) for each cell population. (A) Principle component analysis (PCA) based on differentially expressed (DE) genes of the HSC population; (B) Volcano plot showing differential expressed genes between patients with CAD and individuals without atherosclerosis, controlled for age, in a combined analysis of HSC, MPP, and GMP population. Genes with an FDR <0.1 are named; (C) Gene ontology enrichment analysis of DE genes from HSCs, MPPs, and GMPs, depicting the FDR and enrichment ratio.

The online version of this article includes the following source data and figure supplement(s) for figure 6:

**Source data 1.** Contains source data for *Figure 6B*, and *Figure 6—figure supplements 1B*, *2*, *3*.
**Source data 2.** Contains source data for *Figure 6C*.
**Figure supplement 1.** Transcriptome of HSC, MPP, and GMP populations.
**Figure supplement 2.** Combined heatmap showing the top 50 DE genes for the HSC, MPP, and GMP cell populations in the patients and the control subjects.
**Figure supplement 3.** Separated heatmap showing the top 50 DE genes for each of the HSC, MPP, and GMP cell populations in the patients and the control subjects.

fed mice (*Christ et al., 2018*). Patients with hypercholesterolemia also have circulating monocytes with augmented cytokine production capacity, increased glycolytic metabolism, and enrichment of activating histone modifications, which persist for three months despite cholesterol lowering with statin therapy. Circulating monocyte and neutrophil numbers, however, were not increased in these patients (*Bekkering et al., 2019*). In our current patient cohort, it remains to be established whether trained immunity is present since we did not assess epigenetic markers in the HSPCs.

The observation that absolute numbers of myeloid progenitors and circulating monocytes and neutrophils are not increased, despite functional and transcriptional inflammatory programming, fits with the effects of alternative inducers of trained immunity in humans. Vaccination with the Bacille Calmette-Guérin (BCG) vaccine induces trained immunity in humans *in vivo* (*Arts et al., 2018*). This is accompanied by an increased cytokine production capacity of bone marrow mononuclear cells,

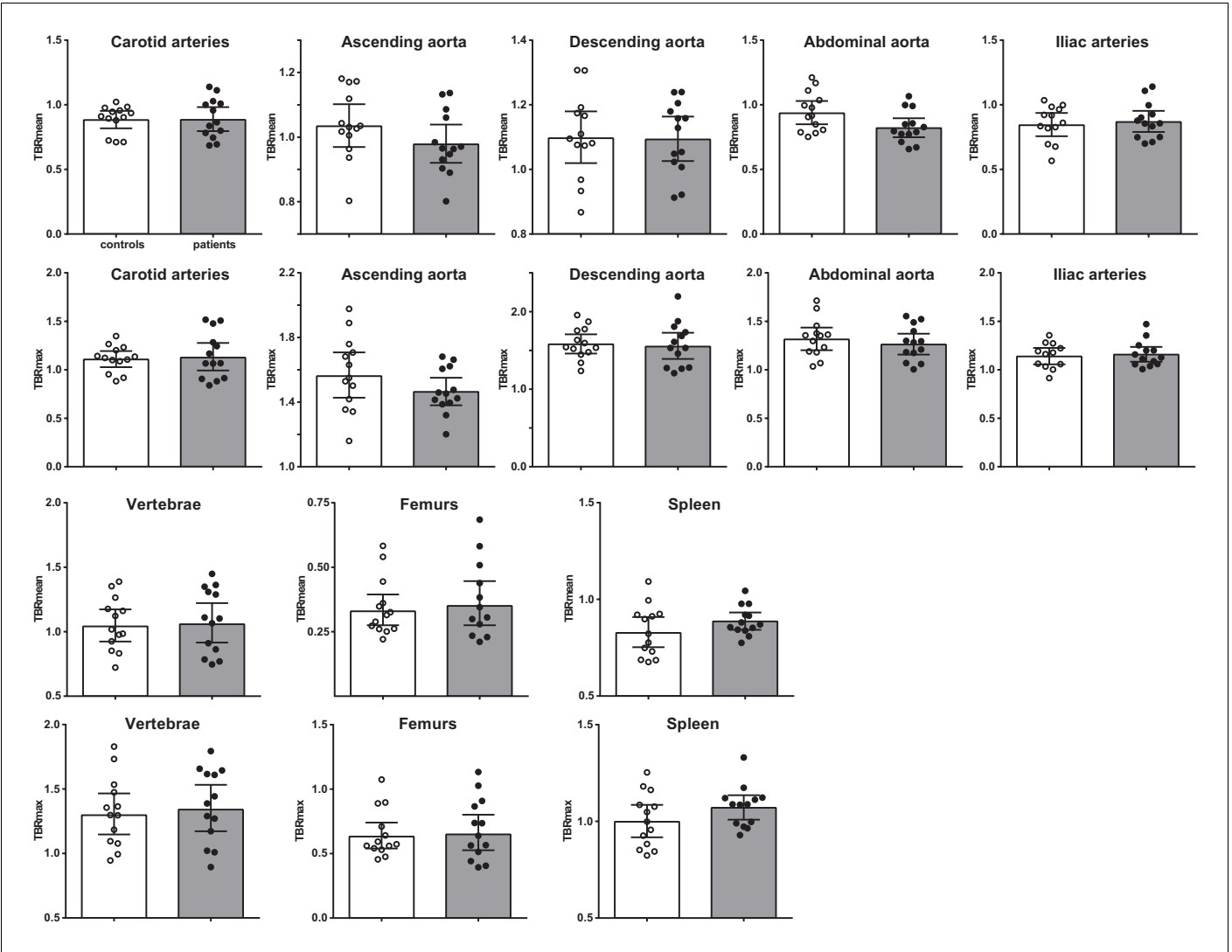

**Figure 7.** Vascular wall inflammation and hematopoietic tissue activation on [$^{18}$F]FDG PET/CT scan. Standard uptake value of each region in control individuals (white bars, n = 13) and individuals with CAD (gray bars, n = 13). Geometric mean with 95% CI. The p-values are corrected for age and BMI with ANCOVA.* indicates p<0.05, **: p<0.01. TBR: target SUV/mean blood pool SUV or mean liver SUV as background.

The online version of this article includes the following figure supplement(s) for figure 7:

**Figure supplement 1.** Splenic activity correlates with progenitor cells and circulating immune cells.

comparable to the result in patients with CAD in the current study (*Cirovic et al., 2020*). Also, 90 days post-BCG administration, HSPCs showed myeloid skewing with activation of neutrophil-associated pathways and pathways associated with regulation of immune responses, comparable to our patients. This functional and transcriptional reprogramming was accompanied by an increase in numbers of circulating myeloid cells in large cohorts of infants vaccinated with BCG (*Cirovic et al., 2020*).

In the patient group with CAD, the increased cytokine production capacity of the bone marrow mononuclear cell fraction was accompanied by an upregulation of both glycolytic metabolism as well as oxygen consumption by oxidative phosphorylation. It is important to realize that the bone marrow mononuclear fraction also contains large amounts of mature monocytes, so we cannot conclude on the metabolic landscape of HSPCs and specific progenitor subtypes. Isolated monocytes and monocyte-derived macrophages from patients with atherosclerosis are characterized by increased glycolysis, which is coupled to the hyperresponsiveness in terms of cytokine release (*Bekkering et al.,*

2016; *Shirai et al., 2016*). Also, HSPCs of mice after induction of trained immunity with β-glucan have an upregulation of the glycolytic pathway (*Mitroulis et al., 2018*).

Analysis of the differentially upregulated genes transcriptome of the flow-sorted HSCs and GMPs showed significant enrichment for neutrophil activation pathways. This further supports a role for neutrophils in the pathophysiology of atherosclerotic CVD, which is suggested by various recent findings in the literature. Although neutrophils are the most abundant circulating white blood cell type, neutrophils only recently received attention in the context of cardiovascular inflammation. A large epidemiological study shows that circulating granulocyte counts are strongly associated with the occurrence of CVD (*Fani et al., 2020*). During atherogenesis, platelet-derived chemokines, such as CC-chemokine ligand 5, promote neutrophil activation and recruitment (*Silvestre-Roig et al., 2020*). At the luminal side, these activated neutrophils secrete granule proteins, including cathepsin G, which further promotes myeloid cell recruitment. Several actions further fuel atherosclerotic plaque formation, including secretion of reactive oxygen species and myeloperoxidase, which mediates oxidation of LDL, promoting foam cell formation (*Silvestre-Roig et al., 2020*). Also, neutrophils can form neutrophil extracellular traps (NETs) (*Döring et al., 2020*) and they can secrete pro-inflammatory microvesicles, which further fuel vascular wall inflammation and atherosclerosis (*Gomez et al., 2020*). Interestingly, a recent study used plasma proteomics analysis to explore the mechanism by which colchicine lowers cardiovascular risk in patients with CAD and showed mainly a downregulation of neutrophil activation pathways (*Opstal et al., 2020*).

Analysis of the individual significantly differentially expressed genes in the HSPCs showed that the glycolytic enzyme phosphofructokinase (*PFKP)* was downregulated in patients with CAD. Quiescent HSCs depend mainly on anaerobic glycolysis to sustain survival, quiescence, and retention within the bone marrow (*Takubo et al., 2010*). Once stimulated to divide, HSCs start to activate OXPHOS to meet metabolic demands of proliferation and differentiation (*Karigane and Takubo, 2017*). Therefore, the downregulation of *PFKP* might be related to the activation of HSCs. This finding is in contrast to the higher extracellular acidification rate of the bone marrow mononuclear cells which points to the activation of glycolysis. We speculate that this finding is dominated by the inflammatory activation of bone marrow monocytes, which constitute a large percentage of the bone marrow mononuclear cells. The chemokine receptor CCR2 showed significant upregulation in the HSPCs of the patients with CAD. CCR2 is critical for monocyte egress from the bone marrow and for the recruitment of circulating monocytes to the arterial wall (*Tsou et al., 2007*). The upregulation of CCR2 in our study might contribute to the increased circulating HSC percentage in patients with CAD. In mice, increased levels of circulating HSPCs are found after myocardial infarction that migrate to the spleen for extramedullary hematopoiesis (*Dutta et al., 2012*). In humans, splenic extramedullary hematopoiesis can occur in response to extreme physiologic stress (e.g. sepsis or hematologic malignancies), but has not been described yet during the development of CVD. Indirect evidence shows that splenic activity in humans is increased after myocardial infarction assessed with PET/CT scanning (*van der Valk et al., 2016*), suggesting increased immune cell activation. In our study, we observed a correlation between splenic activity and CCR2 expression on circulating HSPCs which might suggest that the spleen performs a role in human atherosclerosis.

Our study has several limitations. First, the sample size is relatively small. However, this sample size needs to be considered in the context of the invasive and complex nature of the study, and this sample size usually allows to detect relevant differences (*Cirovic et al., 2020*; *van der Heijden et al., 2020*). The sample size might also be an explanation for the lack of a significant difference in FDG uptake in the large arteries, which has been shown previously in patients with CVD. (*Tarkin et al., 2014*; *van Wijk et al., 2014*). Another possible explanation for our results is that our patients had severe mature coronary atherosclerotic plaques with calcium deposits, which are known to have less FDG uptake than uncalcified arterial plaques at the early stage of atherosclerosis (*Fernández-Friera et al., 2019*). In addition, lipid-lowering therapy is effective in reducing vascular wall inflammation (*Pirro et al., 2019*). Second, we did not explore epigenetic programs of the HSPCs. As such, we cannot conclude whether the mechanism of trained immunity contributes to our findings. This should be the focus of future studies. Third, in a study on symptomatic patients with CVD, it is inevitable to have differences in medication use compared to the healthy control group.

To summarize, our study is the first to prospectively assess HSPC phenotype in patients with CAD and is unique in providing a comprehensive phenotype by the integration of powerful tools such as RNAseq on flow-sorted HSPC populations, functional parameters, Seahorse respiratory, and PET

imaging of vascular wall and hematopoietic activity. These results show that in patients with CAD, activation of the innate immune system occurs at the level of bone marrow myeloid progenitors, which adds exciting opportunities for novel treatment strategies.

# Materials and methods

## Key resources table

| Reagent type (species) or resource | Designation | Source or reference | Identifiers | Additional information |
|---|---|---|---|---|
| Biological sample (*Homo sapiens*) | Peripheral blood | Through venous puncture | | Freshly isolated from *Homo sapiens*, men, 18–75 years |
| Biological sample (*Homo sapiens*) | Bone Marrow aspirate | From the posterior iliac crest according to standard practice | | Freshly isolated from *Homo sapiens* |
| Antibody | Mouse monoclonal CD45 KO | Beckman Coulter | Clone J33 Cat# B36294, RRID:AB_2833027 | (1:25) |
| Antibody | Mouse monoclonal HLA-DR PE | Beckman Coulter | Clone immu-357 Cat# IM1639U RRID:AB_2876782 | (1:10) |
| Antibody | Mouse monoclonal CD14 PECy7 | eBioscience | Clone 61D3 Cat# 25-0149-42 RRID:AB_1582276 | (1:25) |
| Antibody | Mouse monoclonal CD16 FITC | eBioscience | Clone CB16 Cat# 11-0168-42 RRID:AB_10805747 | (1:25) |
| Antibody | Mouse monoclonal CD3 APC-Alexa750 | Beckman Coulter | Clone UCTH1 Cat# A66329 RRID:AB_2876783 | (1:25) |
| Antibody | Mouse monoclonal CD56 APC | Beckman Coulter | Clone N901 Cat# IM2474U RRID:AB_2876784 | (1:25) |
| Antibody | Mouse monoclonal CD192 BV421 | BD Biosciences | Clone 48607 Cat# 564067, RRID:AB_2738573 | (1:50) |
| Antibody | Mouse monoclonal CD11b BV785 | Biolegend | Clone ICRF44 Cat# 301346, RRID:AB_2563794 | (1:50) |
| Antibody | Mouse monoclonal CD41 PerCP-Cy5.5 | Biolegend | Clone Hip8 Cat# 303719, RRID:AB_2561731 | (1:50) |
| Antibody | Mouse monoclonal CD90 FITC | Biolegend | Clone 5E10 Cat# 328107, RRID:AB_893438 | (1:50) |
| Antibody | Mouse monoclonal CD123 PE | BD Biosciences | Clone 9F5 Cat# 555644, RRID:AB_396001 | (1:40) |
| Antibody | Mouse monoclonal CD19 ECD | Beckman Coulter | Clone J4.119 Cat# IM2708U, RRID:AB_130854 | (1:20) |
| Antibody | Mouse monoclonal CD38 PC5.5 | Beckman Coulter | Clone LS198-4-3 Cat# IM2651U, RRID:AB_131166 | (1:20) |
| Antibody | Mouse monoclonal CD117 PEC7 | Beckman Coulter | Clone 104D2D1 Cat# IM3698, RRID:AB_131184 | (1:20) |

*Continued on next page*

*Continued*

| Reagent type (species) or resource | Designation | Source or reference | Identifiers | Additional information |
|---|---|---|---|---|
| Antibody | Mouse monoclonal CD45RA APC | Beckman Coulter | Clone 2H4LDH11LD89 (2H4) Cat# B14807 RRID:AB_2876787 | (1:20) |
| Antibody | Mouse monoclonal CD34-APC A750 | Beckman Coulter | Clone 581 Cat# A89309 RRID:AB_2876786 | (1:20) |
| Commercial assay or kit | DRAQ7 | Biostatus | Live/Dead stain | (1:500) |
| Commercial assay or kit | Human Cytokine Magnetic Magpix 25-plex panel | Invitrogen | | MAGPIX platform |
| Commercial assay or kit | SimplePlex cartridge | ProteinSimple | | Ella platform |
| Commercial assay or kit | Truseq small RNA primers | Illumina | | |
| Commercial assay or kit | hsCRP ELISA | R&D | DY1707 | |
| Commercial assay or kit | VCAM-1 ELISA | R&D | DY809 | |
| Commercial assay or kit | MMP2 ELISA | R&D | DY902 | |
| Commercial assay or kit | E-selectin ELISA | R&D | DY724 | |
| Chemical compound, drug | Pharm Lyse lysing buffer | BD Biosciences | | |
| Chemical compound, drug | Glutamine | Invitrogen | 2 mmol/L in RPMI | |
| Chemical compound, drug | Gentamycin | Centrafarm | 10 mg/mL in RPMI | |
| Chemical compound, drug | Pyruvate | Invitrogen | 1 mmol/L in RPMI | |
| Chemical compound, drug | Methocult GF | Stemcell Technologies | H84435 | |
| Sequence-based reagent | Hg19 human Refseq transcriptome | *Li and Durbin, 2010* | | To align RNAseq |
| Peptide, recombinant protein | Lipopolysaccharide from *Escherichia coli* | Sigma-Aldrich | Serotype 055:B5, L2880 | 10 ng/mL |
| Peptide, recombinant protein | Pam3CysK4 | EMC Microcollections | L2000 | 10 ug/mL |
| Peptide, recombinant protein | Interferon gamma | Immukine, Boehringer Ingelheim BV | | 50 ng/mL for Seahorse |
| Peptide, recombinant protein | Oligomycin | Sigma-Aldrich | 75351 | 1 mM for Seahorse |

*Continued on next page*

*Continued*

| Reagent type (species) or resource | Designation | Source or reference | Identifiers | Additional information |
|---|---|---|---|---|
| Peptide, recombinant protein | FCCP | Sigma-Aldrich | C2920 | 1 mM for Seahorse |
| Peptide, recombinant protein | Rotenone | Sigma-Aldrich | R8875 | 1.25 mM |
| Peptide, recombinant protein | Antimycin A | Sigma-Aldrich | A8674 | 2.5 mM |
| Software, algorithm | Kaluza | Beckman Coulter | Version 2.1 RRID:SCR_016182 | Flow Cytometry analysis |
| Software, algorithm | MultiQC | *Ewels et al., 2016* | RRID:SCR_014982 | Quality check RNAseq |
| Software, algorithm | DEseq2 v1.22.0 | *Love et al., 2014* BioConductor | RRID:SCR_015687 | Differential gene expression RNAseq |
| Software, algorithm | clusterProfiler v3.10.1 | *Yu et al., 2012* BioConductor | RRID:SCR_016884 | RNAseq |
| Software, algorithm | R 3.6.1 | https://www.r-project.org/ | RRID:SCR_001905 | |
| Software, algorithm | TrueX algorithm | EARL protocols | | |
| Software, algorithm | Inveon Research Workspace 4.2 | Preclinical Solutions, Siemens Medical Solutions | 3D Gaussian filter kernel, 3.0 mm | Postprocessing of FDG PET CT scanning |
| Software, algorithm | PyRadiomics toolbox | *van Griethuysen et al., 2017* | | Analysis FDG PET CT |
| Software, algorithm | SPSS V25.0 | SPSS | RRID:SCR_002865 | Data analysis |
| Software, algorithm | Prism v6.0 | GraphPad software | RRID:SCR_002798 | Figures |
| Other | Sysmex-XN 450 hematology analyzer | Sysmex | | For total blood counts |
| Other | CytoFLEX flow cytometer | Beckman Coulter | 13 color on CytExpert RRID:SCR_017217 | Flow Cytometry Peripheral blood |
| Other | Navios flow cytometer | Beckman Coulter | RRID:SCR_014421 | Flow Cytometry Bone marrow Progenitors |
| Other | XFp Analyzer | Seahorse Bioscience | | |
| Other | BD FACSAria II SORP | Becton Dickinson | RRID:SCR_018091 | Flow cytometry sorting |
| Other | Illumina Nextseq500 platform | Illumina | RRID:SCR_014983 | RNAseq |
| Other | Biograph 40 mCT scanner | Siemens Healthineers | ~2.1 MGq/kg FDG i.v. | FDG PET CT |

## Participant selection

Participants (male, 18–75 years) were recruited among patients who were admitted for the evaluation of chest pain and underwent cardiac imaging at the Cardiology department of the Canisius Wilhelmina hospital or the Radboud University Medical Centre, Nijmegen, The Netherlands after January 1, 2015. Severe coronary atherosclerosis was defined as a calcium score >400 HU on computed tomography (CT), or a total plaque score (TPS) >4 on coronary CT angiography (CCTA),

according to previously described standards (*Bekkering et al., 2016*; *Pen et al., 2013*). Control participants, with a calcium score and TPS of zero, were matched for age, body mass index (BMI), and smoking.

Criteria for exclusion were previous cardiovascular events, malignancies, auto-immune or auto-inflammatory diseases (including diabetes mellitus), chronic immunomodulatory drug use, chronic kidney disease (MDRD <45 mL/min), liver disease (ALAT > 135 U/L), or thrombocytopenia (<50 × 10$^6$/mL). Additionally, participants were excluded if they had an infection (>38,5°C or antibiotic treatment), hospital admission, or vaccination within 3 months before study entry. The study protocol was approved by the Institutional Review Board Arnhem/Nijmegen, the Netherlands, and registered at ClinicalTrials.gov (NCT03172507). All individuals gave written informed consent.

## Evaluation of atherosclerotic burden

CCTA total plaque score was calculated as previously described (*Pen et al., 2013*). Briefly, a 64-slice MDCT scanner (Philips) was used to obtain CAC and MDCT Image acquisition using an ECG-synchronized axial scan protocol and post-processing CT and CAC studies using IntelliSpace Philips software. Before image acquisition, beta-blockers were administered targeting a heart rate of <60 beats per minute, and patients received nitroglycerin 0.8 mg sublingually. Prospective electrocardiographically gated step-and-shoot contrast-enhanced MDCT imaging was performed, initiated from 10 mm above the level of the left main artery to 10 mm below the inferior myocardial apex with scan parameters being 64 × 0.625 mm sections (2.5 mm), collimation tube currents of 350 to 780 mAs and tube voltage of 100 or 120 kV. In the rare event that prospective scanning was not possible, retrospective or helix scanning was used.

Reconstruction of the MDCT scans was performed with reconstructed images obtained, using an ECG-triggered protocol, at 75% from the previous RR-interval, or at 75% and 40% from the previous RR-interval if a helix scan-protocol was used. The TPS was determined by summing the number of evaluable coronary segments with calcific or non-calcific plaque, or mixed plaque, where non-calcified and mixed plaque was assigned with one point and calcified plaque with 0 (maximum score = 16). Two independent experienced operators scored all CT images and both operators were blinded for all clinical information. In case of disagreement, the opinion of a third independent observer was asked.

## Study design

Participants underwent [$^{18}$F]FDG PET/CT scanning, follow by venepuncture and bone marrow aspiration within 2–14 days. The participants were invited in pairs (1 patient:1 control) for blood sampling and bone marrow aspiration from December 2017 till July 2018.

## Cardiovascular risk assessment

Medical history, smoking status, medication use, BMI, and fasting glucose concentrations were obtained from all individuals. Blood pressure was measured three times by a manual sphygmomanometer after 5 min seated rest according to AHA guidelines. Fasting total cholesterol (Tchol), high-density lipoprotein cholesterol (HDLc), and triglycerides (TG) were measured using standardized methods, and low-density lipoprotein cholesterol (LDLc) was calculated with the Friedewald formula.

## Blood sampling and bone marrow aspiration

Bone marrow was aspirated from the posterior iliac crest according to standard practice by an experienced physician assistant. Blood was sampled through venous puncture. Sample collection was performed at 8.00–10.00 to avoid interference of circadian rhythms of immune parameters, and sample processing occurred within 2 hours. Plasma and serum were stored at −80°C until further use. Total blood cell counts were determined with an automated Sysmex-XN 450 hematology analyzer (Sysmex, Hamburg, Germany).

## Mononuclear cell enrichment and stimulation

Before mononuclear cell (MNC) enrichment, the bone marrow aspirate was filtered and washed with sterile PBS. Thereafter, the same procedures were followed for peripheral blood MNCs (PBMCs) and bone marrow MNCs (BM-MNCs). PBMCs/BM-MNCs were isolated by Ficoll-Paque density gradient

centrifugation (GE Healthcare, Chicago, IL). Cell composition was evaluated by Sysmex analyzer (Sysmex) and with flow cytometry (*Table 3*, see key resource table (KRT) for RRIDs). PBMCs/BM-MNCs were resuspended in Roswell Park Memorial Institute 1640 Dutch-modified culture medium (RPMI) (Life Technologies/Invitrogen, Waltham, USA) supplemented with 2 mmol/L glutamine (Invitrogen), 10 mg/mL gentamicin (Centrafarm, Etten-Leur, The Netherlands) and 1 mmol/L pyruvate (Invitrogen). Per well, $5 \times 10^5$ PBMCs/BM-MNCs were stimulated for 24 hours in duplicate in round-bottom 96-well plates (Corning, NY) with the following stimuli: RPMI, 10 ng/mL *Escherichia coli* lipopolysaccharide (LPS) (serotype 055:B5 Sigma-Aldrich, St. Louis, MO), and 10 µg/mL Pam3CysK4 (P3C) (EMC Microcollections, Tübingen, Germany). After 24-hour incubation, supernatants were stored after plate centrifugation at −80°C until cytokine assessment.

## Cytokine measurements

Cytokine and chemokine concentrations were determined in supernatants using Human Cytokine Magnetic Magpix 25-plex panel (Invitrogen) on the MAGPIX platform (Luminex, Austin, TX). Circulating IL-1β, IL-1Ra, IL-6, and IL-18 concentrations were measured with the SimplePlex cartridge on the Ella platform (ProteinSimple, San Jose, CA). Additional circulating cytokines/chemokines concentrations were measured using ELISA (see KRT).

## Flow cytometry

In the circulation, monocyte subpopulations and expression markers were determined with flow cytometry. 50 µL EDTA blood was stained after the lysis-no-wash strategy (BD Pharm Lyse lysing buffer, Becton Dickinson) by monoclonal antibodies CD45 Krome Orange ([KO], clone J33; Beckman Coulter, Cat# B36294, RRID:AB_2833027), HLA-DR PE (clone immu-357; Beckman Coulter, Cat# IM1639U, RRID:AB_2876782), CD14 PC7 (clone 61D3e Bioscience, Cat# 25-0149-42, RRID:AB_1582276), CD16 FITC (clone CB16; eBioscience Cat# 11-0168-42, RRID:AB_10805747), CD3 APC-Alexa750 (clone UCTH1; Beckman Coulter, Cat# AA66329, RRID:AB_2876783), CD56 APC (clone N901; Beckman Coulter, Cat# IM2474U, RRID:AB_2876784), CD192 Brilliant Violet421 ([BV421] clone 48607; Becton Dickinson, Cat#564067, RRID:AB_2738573), CD11b BV785 (clone ICRF44; Biolegend, Cat#301346, RRID:AB_2563794), CD41 PC5.5 (clone Hip8; Biolegend, Cat# 303719, RRID:AB_2561731) and measured with CytoFLEX flow cytometer (Beckman Coulter, RRID:SCR_017217). The gating strategy applied is shown in *Supplementary file 1*, gates were set with the fluorescence-minus-one method (*Weber et al., 2016*; *Ziegler-Heitbrock et al., 2010*). In short, monocytes were selected based on CD45+ HLA-DR+ and monocyte scatter properties, then CD3+ T-lymphocytes and CD56+ NK-cells were excluded, and monocyte subsets were identified in the CD14/CD16 plot as percentage of gated. Data was analyzed with Kaluza 3.1 software (Beckman Coulter, RRID:SCR_016182). Characterization of monocytes subsets is according to current recommendations (*Weber et al., 2016*; *Ziegler-Heitbrock et al., 2010*).

Bone marrow progenitors were identified with Navios flow cytometer (Beckman Coulter, RRID:SCR_014421). $5 \times 10^6$ Bone marrow cells were washed, lysed, and stained for 20 min with the monoclonal antibodies: CD90 FITC (Clone 5E10; Biolegend Cat#328107, RRID:AB_893438), CD123 PE (Clone 9F5; BD Biosciences, Cat#555644, RRID:AB_396001), CD19 ECD (Clone J3.119; Beckman Coulter, Cat# IM2708U, RRID:AB_130854), CD38 PC5.5 (Clone LS198-4-3; Beckman Coulter, Cat# IM2651U, RRID:AB_131166), CD117 PEC7 (Clone 104D2D1; Beckman Coulter, Cat# IM3698, RRID:

**Table 3.** Cell composition of PBMC and BM-MNC fraction.

| Cell types in PBMC fraction | Controls | Patients |
|---|---|---|
| Lymphocytes (%) | 73 [68-79] | 65 [62-75]* |
| Monocytes (%) | 25 [19-31] | 32 [23-35] |
| Neutrophils (%) | 0.7 [0.6–1.1] | 1.2 [0.6–1.7] |
| Cell types in BM-MNC fraction | | |
| HSPCs (%) | 1.6 [1.2–2.0] | 1.4 [1.2–4.7] |

Cellular composition after mononuclear cell enrichment of peripheral blood and bone marrow. Median with [IQR]. Mann-Whitney U test. *: p<0.05, **: p<0.01. HSPCs: hematopoietic stem and progenitor cells.

AB_131184), CD45RA APC (Clone 2H4LDH11LD89; Beckman Coulter, Cat# B14807, RRID:AB_2876787), DRAQ7 (Biostatus), CD34-APC A750 (Clone 581; Beckman Coulter, Cat# A89309, RRID: AB_2876786), CD192 PB (Clone 48607; Becton Dickinson, Cat# 564067, RRID:AB_2738573), CD45 KO (Clone J33; Beckman Coulter, Cat# B36294, RRID:AB_2833027). The target input was set at $1 \times 10^6$ measured events, panels were balanced with the fluorescence-minus-one method to determine spill over, spectral overlap, and nonspecific binding.

The gating strategy is displayed in *Figure 2—figure supplement 1*. In short, HSPCs were defined as CD45+CD34+CD38dim/+ cells, after selecting for singlets and alive cells. Next, the lymphoid lineage was excluded by the gating of CD19-CD117+ cells. In CD45RAdimCD38+ gated cells, CMP, GMP, MEP, and R1-3 progenitor populations were identified using CD123 and CD45RA expression. CD90 expression in CD38-CD45RA- cells determined MPP and HSC populations. Data were analyzed with Kaluza 2.1 software.

## Proliferation assays

$2.5 \times 10^4$ BM-MNCs were cultured in methylcellulose medium containing erythropoietin, G(M)-CSF, IL-3 and IL-6 (MethoCult GF H84435, Stemcell technologies, Vancouver, Canada) and 2% fetal bovine serum (Integro B.V., Zaandam, The Netherlands) in Petri dishes for 14 days in 37°C 5% $CO_2$ incubator. After 14 days, erythroid progenitor (BFU-E), common myeloid progenitor (CFU-GEMM), and granulocyte-macrophage progenitor (CFU-GM) cell colonies were counted in duplo by an experienced hematology operator.

## Mitochondrial respiration and glycolysis assays

Using a modified protocol from *Shirai et al., 2016*, $2 \times 10^6$ BM-MNCs were plated per well in fivefold to overnight-calibrated cartridges in assay medium (DMEM with 1 mM L-Glutamine, 11 mM glucose, and 1 mM pyruvate [pH adjusted to 7.4]) and incubated for 1 hr in a non-$CO_2$-corrected incubator at 37°C. In addition, BM-MNCs were stimulated with 50 ng/mL interferon gamma (IFN-γ) (Immukine, Boehringer Ingelheim BV, Alkmaar, The Netherlands) and 10 ng/mL LPS (Sigma-Aldrich) for 2 hr. Oxygen consumption rate (OCR) and extracellular acidification rate (ECAR) were measured via XFp Analyzer (Seahorse Bioscience, North Billerica, MA), with final concentrations of 1 mM oligomycin, 1 mM FCCP, and the combination of 1.25 mM rotenone and 2.5 mM antimycin A.

## RNA sequencing of sorted progenitor populations and circulating monocytes

HSPCs were sorted by BD FACSAria II SORP flow cytometer sorter (Becton Dickinson, RRID:SCR_018091), using the flow cytometric panel as described above (*Figure 2—figure supplement 1*). BM-MNCs stored in liquid nitrogen were gently thawed in fetal calf serum (in house) containing 1.25 mM MgCl2 and 0.1 mg/mL DNase I (Sigma). HSCs, MPPs, and GMPs populations were directly sorted in 100 μL TRIzol (ThermoFisher, Waltham, MA) before processing for RNA sequencing.

Total RNA was extracted using the standard TRIzol (ThermoFisher) protocol and used for library preparation and sequencing. mRNA was processed as described previously, following an adapted version of the single-cell mRNA seq protocol of CEL-Seq (*Hashimshony et al., 2012*; *Simmini et al., 2014*). In brief, samples were barcoded with CEL-seq primers during reverse transcription and pooled after second strand synthesis. The resulting cDNA was amplified with an overnight i*n vitro* transcription reaction. From this amplified RNA, sequencing libraries were prepared with Illumina Truseq small RNA primers (Illumina, San Diego, CA). Paired-end sequencing was performed on the Illumina Nextseq500 platform (RRID:SCR_014983). Read one was used to identify the Illumina library index and CEL-Seq sample barcode. Read two was aligned to the hg19 human RefSeq transcriptome using BWA (*Li and Durbin, 2010*). Reads that mapped equally well to multiple locations were discarded. RNA input for all samples was normalized, and libraries for each progenitor population were sequenced in a single run.

## Bioinformatics, differential gene expression, and pathway analysis

Reads were mapped to hg19 human reference genome using BWA (*Li and Durbin, 2010*). MultiQC (RRID:SCR_014982) was used to quality check all the samples (*Ewels et al., 2016*). In total, 107,565,838 reads were mapped in 59 progenitor populations, with one GMP population failed to

pass quality control. Raw read counts provided the input for differential gene expression analysis with DESeq2 (RRID:SCR_015687) (*Love et al., 2014*) after excluding genes with less than 50 reads. Internal variance stabilizing transformation was used for regressing out age and to normalize the data (Benjamini–Hochberg correction was used to adjust p-value for multiple testing correction). In the differential expression analyses within HSC populations, one HSC population was marked as an outlier and further filtered out as it located three standard deviations away from mean values in principle component analysis of this study.

Differential gene expression analysis was carried out after regressing out age with DESeq2 v1.22.0 in R (RRID:SCR_001905) (*Love et al., 2014*), with internal statistical and normalization method (i.e. correction for multiple testing with Benjamini–Hochberg). The average expression between individuals with atherosclerosis (n = 10) and matched individuals without atherosclerosis (n = 10) was analyzed for each progenitor population. Enrichment analysis was performed using R package clusterProfiler (RRID:SCR_016884) (*Yu et al., 2012*) with Gene Ontology terms (*Ashburner et al., 2000*). After BH adjustment, a FDR <0.05 in enrichment analysis was considered significant.

## [$^{18}$F]FDG PET and low-dose CT scanning

Participants underwent [$^{18}$F]FDG PET with low-dose non-contrast-enhanced CT from skull base to the trochanter major on a dedicated Siemens Biograph 40 mCT scanner (Siemens Healthineers, Erlangen, Germany). After adhering to a 24-hour low-carbohydrate diet and 6 hours of fasting,~2.1 MBq/kg [$^{18}$F]FDG was administered intravenously, as described previously (*Bucerius et al., 2016*). Glucose concentrations were obtained (5.3 ± 0.5 mmol/L) after injection. Before scanning, participants rested in the supine position for 2 hours.

Images were reconstructed according to EARL protocols; using a TrueX algorithm with point spread function (PSF) and time-of-flight (TOF) measurements, using three iterations, 21 subsets, matrix size 200 × 200 (pixel spacing of 4.07 mm), full width half maximum (FWHM) of 3 mm and using 2 min of PET data. Postprocessing was performed using a 3D Gaussian filter kernel, 3.0 mm, using the Inveon Research Workspace 4.2 (Preclinical Solutions, Siemens Medical Solutions USA, Knoxville, TN).

[$^{18}$F]FDG-uptake in the vascular wall was determined in seven regions of interest (ROI) by a single investigator (MPN); the aorta ascendens, aorta descendens, abdominal aorta, the left and right common carotid arteries, and the left and right iliac arteries. [$^{18}$F]FDG-uptake in hematopoietic tissue was assessed in the spleen, lumbal vertebrae L2 and L3, and in the left and right medullary bone of the femur. These regions of interest were evaluated using the Inveon Research Workspace 4.2. The standardized uptake value (SUV) was extracted from each ROI after correction for [$^{18}$F]FDG dose (MBq) and BMI using the PyRadiomics toolbox (*van Griethuysen et al., 2017*). The SUVs of left and right ROIs (e.g. left and right carotid artery) were averaged. Next, the target-to-background ratio (TBR) was calculated as the ratio of the vascular wall SUV and the mean thoracic arterial blood pool SUV. The TBRs of hematopoietic tissue were expressed as the ratio of the mean liver SUV. The primary outcome is the TBR, as recommended by the European guideline (*Bucerius et al., 2016*).

## Statistical analysis

This study is exploratory, hence no sample size calculation is performed. Normal distribution of the data was checked with the Shapiro-Wilk test, when the p-value reached <0.05 this assumption was violated and non-parametric tests were used. Data are reported as mean ± SD with independent samples T-test according to Levene's test for equality of variances, as mean (number of participants) with $X^2$ test for categorical data, and as median [interquartile range] with Mann-Whitney U test for non-parametrical data. Outliers were removed with a standard deviation >± 2.5 of Z-scores. All outcomes were log(10)-transformed and thereafter corrected with ANCOVA for confounding demographics such as age and BMI. SPSS V25.0 (SPSS Inc, Chicago, IL, RRID:SCR_002865) and Graphpad Prism v6.0 (GraphPad software, La Jolla, CA, RRID:SCR_002798) were used for data analysis and visualization. A two-sided p-value<0.05 was considered statistically significant.

## Acknowledgements

We would like to thank Cor Jacobs for performing the flow cytometric measurements, and Rob Woestenenk for his expertise with flow sorting at the Radboud technology center flow cytometry.

## Additional information

### Funding

| Funder | Grant reference number | Author |
|---|---|---|
| Horizon 2020 | 667837 | Leo AB Joosten<br>Mihai G Netea<br>Niels P Riksen |
| Netherlands Organisation for Scientific Research | NWO SPI 94-212 | Mihai G Netea |
| European Commission | 833247 | Mihai G Netea |
| ERA-NET | 2018T093 | Niels P Riksen |
| Netherlands Organisation for Scientific Research | 452173113 | Siroon Bekkering |
| Hartstichting | 2018T028 | Siroon Bekkering |
| Hartstichting | CVON2018-27 | Leo AB Joosten<br>Mihai G Netea<br>Niels Peter Riksen |

The funders had no role in study design, data collection and interpretation, or the decision to submit the work for publication.

### Author contributions

Marlies P Noz, Data curation, Formal analysis, Investigation, Visualization, Writing - original draft, Project administration; Siroon Bekkering, Conceptualization, Formal analysis, Supervision, Writing - original draft, Writing - review and editing; Laszlo Groh, Tim MJ Nielen, Evert JP Lamfers, Erik HJPG Huys, Formal analysis, Investigation, Writing - review and editing; Andreas Schlitzer, Formal analysis, Supervision, Writing - review and editing; Saloua El Messaoudi, Investigation, Writing - review and editing; Niels van Royen, Supervision, Investigation, Writing - review and editing; Frank WMB Preijers, Conceptualization, Formal analysis, Supervision, Writing - review and editing; Esther MM Smeets, Data curation, Formal analysis, Writing - review and editing; Erik HJG Aarntzen, Marc E Gomes, Conceptualization, Supervision, Writing - review and editing; Bowen Zhang, Data curation, Software, Formal analysis, Visualization, Writing - original draft, Writing - review and editing; Yang Li, Conceptualization, Data curation, Supervision, Methodology, Writing - review and editing; Manita EJ Bremmers, Investigation, Methodology, Writing - review and editing; Walter JFM van der Velden, Harry Dolstra, Supervision, Methodology, Writing - review and editing; Leo AB Joosten, Mihai G Netea, Conceptualization, Supervision, Funding acquisition, Writing - review and editing; Niels P Riksen, Conceptualization, Resources, Supervision, Funding acquisition, Investigation, Methodology, Writing - original draft, Project administration, Writing - review and editing

### Author ORCIDs

Marlies P Noz (iD) https://orcid.org/0000-0002-2299-8313
Siroon Bekkering (iD) http://orcid.org/0000-0003-1149-466X
Tim MJ Nielen (iD) https://orcid.org/0000-0001-7762-5912
Evert JP Lamfers (iD) https://orcid.org/0000-0002-5582-3720
Leo AB Joosten (iD) http://orcid.org/0000-0001-6166-9830
Niels P Riksen (iD) https://orcid.org/0000-0001-9197-8124

## Ethics

Human subjects: Informed consent was obtained for all participants. The study protocol was approved by the Institutional Review Board Arnhem/Nijmegen, the Netherlands and registered at the ClinicalTrials.gov (NCT03172507).

## Decision letter and Author response

Decision letter https://doi.org/10.7554/eLife.60939.sa1
Author response https://doi.org/10.7554/eLife.60939.sa2

## Additional files

### Supplementary files

• Supplementary file 1. Gating strategy of circulating immune cells (related to *Table 2*). Monocytes were selected based on CD45+ HLA-DR+ and monocyte scatter properties, after exclusion of dead cells and doublets. Then CD3+ lymphocytes and CD56+ NK-cells were excluded, and monocyte subsets were identified in the CD14/CD16 plot as the percentage of gated. CD11b and CCR2 expression was determined on the monocyte population.

• Transparent reporting form

### Data availability

RNA-seq data have been deposited in the ArrayExpress database at EMBL-EBI (https://www.ebi.ac.uk/arrayexpress) under accession number E-MTAB-9399.

The following dataset was generated:

| Author(s) | Year | Dataset title | Dataset URL | Database and Identifier |
|---|---|---|---|---|
| Noz M, Bekkering S, Riksen NP, Netea MG | 2020 | Bone marrow progenitor cell populations of patients with severe coronary artery disease | https://www.ebi.ac.uk/arrayexpress/experiments/E-MTAB-9399/ | ArrayExpress, E-MTAB-9399 |

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
