## [Decision Letter]

**Acceptance summary:**

In an in-depth analysis of the transcriptional changes occurring in bone marrow progenitors of patients affected by severe coronary artery disease, Noz et al. demonstrate an increase in vitro cytokine production by peripheral blood mononuclear phagocytes of affected patients upon stimulation with innate ligands. Furthermore, RNA-seq of progenitor cells shows an enrichment in the expression of genes indicative of neutrophil and monocyte activation and differentiation. This study provides the first evidence for bone marrow myeloid progenitor cells are reprogrammed towards myeloid skewing and inflammatory activation in patients with severe coronary artery disease.

**Decision letter after peer review:**

Thank you for submitting your article "Reprogramming of bone marrow myeloid progenitor cells in patients with severe coronary atherosclerosis" for consideration by *eLife*. Your article has been reviewed by three peer reviewers, and the evaluation has been overseen by a Reviewing Editor and a Senior Editor. The following individual involved in review of your submission has agreed to reveal their identity: Roxane Tussiwand (Reviewer #3).

The reviewers have discussed the reviews with one another and the Reviewing Editor has drafted this decision to help you prepare a revised submission.

Summary:

The study by Noz et al. is an in-depth analysis of the transcriptional changes occurring in the bone marrow progenitors of patients affected by severe coronary atherosclerosis. The study collects data from 13 patients with severe coronary atherosclerosis and 13 healthy volunteers. The authors observed an increase in vitro cytokine production by peripheral blood mononuclear phagocytes of affected patients upon stimulation with innate ligands. Furthermore, RNA-seq of progenitor cells shows an enrichment in the expression of genes related to neutrophils and monocyte activation and differentiation.

This is an interesting and well-written paper supported by a large amount of data suggesting that BM derived myeloid progenitor cells are reprogrammed in human atherosclerosis. The study is well-conducted with appropriate statistical approaches suitable for current exploitative study.

Essential revisions:

1) The reviewers understand that the authors investigated "severe coronary atherosclerosis", not "severe coronary disease". However, the reviewers would like to confirm that the recruited patients were free of clinical symptoms related to coronary artery disease including angina.

2) If possible, a cytokine and chemokines array should be performed on the serum of patients. Are neutrophil or myeloid related cytokines and chemokines systemically enriched? i.e. FLT3L, G-CSF, GM-CSF… (the increase in circulating HSCs could reflect increased systemic mobilizing factors)

3) Please include a complete panel of the patient's and donor's PBMC (Table 2). It is important to understand if skewing is observed in the periphery. Trained immunity leads increased myelopoiesis besides increased responsiveness.

4) Before and after the in vitro stimulation include a flow panel on the main subsets. It may be relevant to understand that population skewing can partially explain the differential cytokine expression.

5) Similarly, what is the subset distribution in the BM samples?

6) For the metabolic analysis a subset distribution showed be included before and after cytokine stimulation in parallel to the Seahorse analysis. Moreover, rather than showing an individual data point a time course should be performed using the different inhibitors for the evaluation of spare capacity and glycolytic reserve as it is normally done (OCR treatment with Oligomycin, FCCP and Rotenone/Antimycin; for ECAR Glucose, oligomycin and 2DG).

---

## [Author Response]

Essential revisions:1) The reviewers understand that the authors investigated "severe coronary atherosclerosis", not "severe coronary disease". However, the reviewers would like to confirm that the recruited patients were free of clinical symptoms related to coronary artery disease including angina.

We apologize to the reviewers for not being clear on this point: the patients were all admitted to the cardiac emergency ward for evaluation of chest pain and were subsequently diagnosed with severe coronary atherosclerosis. So these patients suffer from symptomatic coronary atherosclerosis. In the revised version, we have stated this in the Materials and methods section and in the Abstract. In addition, we changed the title into ‘reprogramming of bone marrow myeloid progenitor cells in patients with coronary artery disease’, indicating the symptomatic nature of the atherosclerotic lesions.

2) If possible, a cytokine and chemokines array should be performed on the serum of patients. Are neutrophil or myeloid related cytokines and chemokines systemically enriched? i.e. FLT3L, G-CSF, GM-CSF… (the increase in circulating HSCs could reflect increased systemic mobilizing factors)

We would like to thank the reviewer for this interesting suggestions. We have now performed the additional ELISA’s after purchasing the ELISA kits. For FLT3L we observed a trend towards a higher circulating concentration in the CAD patients compared to the controls, albeit not statistically significant (see Author response image 1; P=0.12).

For the G-CSF and GM-CSF unfortunately all samples except one were below the limit of detection and we cannot draw any conclusions from this. We have chosen not to include these data in the manuscript. However, if the Editor would like them to be incorporated, we will do so.

3) Please include a complete panel of the patient's and donor's PBMC (Table 2). It is important to understand if skewing is observed in the periphery. Trained immunity leads increased myelopoiesis besides increased responsiveness.

All results of the leukocyte differentiation with the Sysmex analyser are shown in Table 2. The results of the flow cytometry are mentioned in the Table 2 for the monocytes and in Figure 2 for the circulating progenitor cells. We do not have any additional data. Table 2 clearly showed that there is no skewing in the circulating PBMC’s. This is consistent with the findings after BCG-induced trained immunity in vivo that we recently published (Cirovic et al., 2020): 90 days after BCG vaccination, circulating monocytes clearly showed a hyperresponsive trained phenotype, but Differential blood count analysis did not reveal significant differences between the BCG-vaccinated and BCG-naive group within neutrophils, total lymphocytes or monocyte counts. In addition, there was no difference within the PBMC fraction in percentages of classical/intermediate/non-classical monocytes, and DC’s.

4) Before and after the in vitro stimulation include a flow panel on the main subsets. It may be relevant to understand that population skewing can partially explain the differential cytokine expression.

The ex vivo stimulation experiments have been performed with isolated PBMCs from the patients and controls. As reported in Table 2, there are no differences in leukocyte differentiation and subsets in whole blood, and therefore the higher cytokine production capacity that we observed is not due the differential skewing. As an additional confirmation that there were no large differences after PBMC isolation, we also performed Sysmex analysis on the PBMC’s, which are shown in Table 3 and confirm no differences between patients and controls. We did not repeat the flow cytometry on the PBMC fractions. Also, we did not perform flow cytometry after the 24 hours of ex vivo stimulation, since the PBMCs are terminally differentiated cells and hence no skewing is to be expected, and the monocytes will be differentiated into macrophages. We hope that this explanation is helpful for the reviewer.

5) Similarly, what is the subset distribution in the BM samples?

For the bone marrow, we performed flow cytometry to identify the various progenitor cell types (HSC, MPP, CLP, CMP, GMP, MEP and pre-monocytes), and the results are shown in Figure 2.

We think that the reviewers now refers to the monocyte subset distribution of the monocytes in the bone marrow and we have added this information in the revised version “Within bone marrow, no difference in the percentage of monocytes (3.3 [3.1-4.4] % in patients versus 3.4 [3.0-4.6] in controls), and monocyte subpopulations, i.e. classical (87 [82-88] % versus 82 [79-89]), intermediate (7.5 [7.1-10.8] % versus 9.3 [6.1-11.5]) and non-classical monocytes (5.1 [4.2-8.1] % versus 8.1 [3.6-11.4]), was observed.”

6) For the metabolic analysis a subset distribution showed be included before and after cytokine stimulation in parallel to the Seahorse analysis. Moreover, rather than showing an individual data point a time course should be performed using the different inhibitors for the evaluation of spare capacity and glycolytic reserve as it is normally done (OCR treatment with Oligomycin, FCCP and Rotenone/Antimycin; for ECAR Glucose, oligomycin and 2DG).

With regard to the first request, we would like to stress that the subset distribution before the Seahorse analysis can be derived from Table 2: this is the baseline leukocyte differentiation in the blood of the patients and controls. We did not perform flow cytometry after the Seahorse measurements.

With regard to the second request: we have now added the time-plots of the Seahorse data with the effect of the different inhibitors indicated (see Figure 4).